# Tomato Crop Performances under Chemical Nutrients Monitored by Electric Signal

**Gabriela Mihalache [1,2], Catalina Iuliana Peres [1], Ilie Bodale [1], Vladut Achitei [1], Madalin Vasile Gheorghitoaie [1], Gabriel Ciprian Teliban [1], Alexandru Cojocaru [1], Monica Butnariu [3,*], Vergil Muraru [4] and Vasile Stoleru [1,*]**

[1] Department of Horticultural Technologies, "Ion Ionescu de la Brad" University of Agricultural Sciences and Veterinary Medicine, 3 M. Sadoveanu, 700440 Iasi, Romania; gabriela.mihalache@uaic.ro (G.M.); peres_catalina@uaiasi.ro (C.I.P.); ilie.bodale@uaiasi.ro (I.B.); achitei.vladut@yahoo.com (V.A.); madalingvm@yahoo.com (M.V.G.); gabrielteliban@uaiasi.ro (G.C.T.); acojocaru@uaiasi.ro (A.C.)

[2] Integrated Center of Environmental Science Studies in the North East Region (CERNESIM), The "Alexandru Ioan Cuza" University of Iasi, 700506 Iasi, Romania

[3] Chemistry & Biochemistry Discipline, Banat's University of Agricultural Sciences and Veterinary Medicine "King Michael I of Romania" from Timisoara, Calea Aradului 119, 300645 Timis, Romania

[4] The National Institute of Research—Development for Machines and Installations Designed to Agriculture and Food Industry, 6 Ion Ionescu de la Brad Street, 010082 Bucharest, Romania; virgil.muraru@gmail.com

**\*** Correspondence: monicabutnariu@yahoo.com (M.B.); vstoleru@uaiasi.ro (V.S.)

**Abstract:** Fertigation is considered an efficient alternative to the enhanced use of chemical fertilizers. Since most of the fertigation systems rely on a theoretical electrical conductivity value of the nutrient solution, we tried to evaluate if this is the real need of the plants as well as if all the nutrients are needed at once. Therefore, we analyzed the electrical signals of the nutrients applied individually or in different mixes, correlating the electrical signals with the leaf gas exchange processes, studying the relation between the electrical signals and different plant phenological stages and the influence of the treatments on the lycopene content, nutritional composition, and antinutritional factors as well as the mineral bioavailability of tomato fruits cv. Brillante F1. The study was carried out in a greenhouse under controlled conditions. Ten different treatments consisting of MaEs (major elements) (V1—$MgSO_4$, V2—$KNO_3$, V3—$K_2SO_4$, V4—$Ca(NO_3)_2$, V5—$KH_2PO_4$, V6—KCl, V7—$MgSO_4 + KNO_3 + KH_2PO_4$, V8—$K_2SO_4 + Ca(NO_3)_2 + KCl$, V9—the mix of V1 to V6, commonly used in agricultural practices, V10—one nutrient each day) were applied daily when plants were 42 days old. The results showed that the values of the electrical signals varied depending on the treatment and the plant phenological stage. Five different trends of the electrical signals were identified. In addition, the shape of the signals varied during the day in accordance with the photosynthesis and the amount of $CO_2$ registered. The results of the treatments' influence on the nutritional composition and lycopene content of tomato fruits suggested that plants do not need all the nutrients at once; the highest values are registered for $K_2SO_4$ fertilization. However, this fertilizer also had the highest registered tannin, saponin, and trypsin inhibitors content, constituting a disadvantage considering the high nutritional values and lycopene content. Regarding the bioavailability of zinc, calcium, and iron for the human diet, regardless of the treatment applied, phytic acid did not affect the availability of zinc and calcium, but it had a negative impact on iron availability; also, the amount of oxalate could impair the bioavailability of calcium. The study suggests that tomato plants do not need all nutrients at once for quality fruits. However, further studies are needed in order to develop a fertigation scheme based on a smart nutrient use that provides an improved nutritional composition and mineral bioavailability. In addition, it is necessary to evaluate the influence of treatments on yield.

**Keywords:** macronutrients; lycopene; chemical fertilization; nutritional value; antinutritional factors

## 1. Introduction

The agricultural land used for vegetable production is an important share of the total area intended for agriculture. Almost 2.2 million hectares of arable land in the European Union is used to grow vegetables for fresh consumption or for processing. Among vegetable crops, tomatoes cover the largest area of the total land used for vegetable production, of about 11.7% [1]. In 2016/2017, the European Union produced more than 18 million tons of tomatoes, which until 2030 is expected to remain relatively the same [2,3]. Tomatoes are the most cost-effective vegetables for growers, with a high versatility for cultivation. Thus, tomatoes can be grown in open fields or in greenhouses, in hydroponic systems, or on different substrates (date–palm waste compost, vermicompost, animal manure) [4–8]. To meet the increased demand for tomatoes, farmers are using intensive amounts of fertilizers [9].

Fertilizers play an important role in providing nutrients to plants and in sustaining an optimal crop yield. In general, plants need three major elements (MaEs) for their optimal growth and development: nitrogen (N), phosphorus (P), and potassium (K). Most of the modern chemical fertilizers contain one or all of these nutrients. Other important elements are sulfur (S), magnesium (Mg), and calcium (Ca). Micronutrients (MiEs) such as iron (Fe), chlorine (Cl), copper (Cu), manganese (Mn), zinc (Zn), molybdenum (Mo), and boron (B) are needed just in small amounts but are equally important for the plants [10]. Along with the increasing population and the need for more food, the fertilizer demand has increased, leading to an excessive application in agriculture. Studies have shown that raising the amount of fertilizers resulted in low crop production and low nutrient-use efficiency, causing environmental problems [11]. The main negative effects of excessive fertilization on the environment are the losses of N through various means (2–20% lost evaporation, 15–25% react organic compounds in the clay soil, 2–10% interfere surface, and ground water) and the losses of P through surface runoff and soil erosion [12,13].

The use of fertilization in a traditional way has led to serious environmental problems consisting in an accumulation of inorganic pollutants with negative impact on water resources, soil, air, and product quality [13]. According to the FAO, soil pollution has been identified as the third major threat to soil functions in Europe and Eurasia. Excessive application of fertilizers combined with an inefficient use of the nutrients by the plant root system contributed to soil degradation and contamination. In the European Economic Area, there are approximately three million potentially polluted sites [11]. Therefore, an effective management of fertilizer usage needs to be done in order to allow a rapid uptake of nutrients by plants and a decrease in accumulation of pollutants.

Fertigation represents an alternative to the enhanced use of fertilizers. Fertigation consists of an application of plant nutrients through a drip or sprinkler irrigation system [10,12,14]. The main advantage of fertigation is the control on the time and rate of water and fertilizers applied in accordance to the crop requirement at every physiological growth stage [14]. Among the results are the increases in nutrients use efficiency by minimizing their losses, savings in fertilizers and water consumption, which can be up to 30–50%, uniform distribution of fertilizers, crop yield increases by 25–33%, reduction of pollutants accumulation, and of course, savings of time, labor, and energy [10,14,15]. Another advantage of fertigation technology is its suitability for both the field and greenhouses cultivation systems. Usually in fertigation, the electrical conductivity (EC) value as well as the pH of the nutrient solution are used in order to establish the optimal amount of fertilizers needed to be add in the water [16]. In addition, electric current measurements provide useful information about crop nutrition level, taking into account that electrical signals reflect the plant responses to different parameters changes [17]. The electrical signal generated and transmitted in plants depends on internal factors, such as osmotic pressure [18], cytoplasmic calcium concentration [19], as well as on external factors such as temperature [20], light [21], water availability in soil [22], and mechanical and electrical stimuli [23]. In addition, it also depends on the physiological processes due to gas exchanges, photosynthesis, or phototropism [24]. There are already studies that have investigated different methods for supplying crops with a specific EC of nutrient solution but the accuracy, the time lag, and stability of the fertigation system still need to be improved [25,26]. Moreover, most of the fertigation systems are based on a

theoretical EC value corresponding to the plants need [27–32]. However, are those EC values of the nutrient solution reflecting the real need of the plants? In addition, do plants need all the nutrients at once? This study is trying to answer these questions by (1) analyzing the electrical signals of the nutrients applied individually or in different mixes; (2) studying the influence of the treatments on the lycopene content, nutritional and antinutrient composition, and the mineral bioavailability of tomato fruits cv. Brillante F1; and (3) correlating the electrical signals with the leaf gas exchange processes, different plant phenological stages, lycopene content, nutritional and antinutrient composition.

## 2. Materials and Methods

### 2.1. Plant Material

"Brillante F1" tomato cultivar was used in the experiment. This cultivar was selected because (1) it is destined for cultivation in protected areas, (2) it is known for its high yield in both conventional and soilless cultivation systems, and (3) it is resistant to *Verticillium* Wilt, *Fusarium* Wilt, and Tomato Mosaic Virus.

### 2.2. Experimental Design

Tomato seeds were first germinated in a growth chamber under controlled conditions (temperature of 22 °C; day length, 10 h–10.000 Lux; relative humidity of 75%). When the seedlings had their first leaf, they were transferred in 400 $cm^3$ plastic pots using Kekkila peat as substrate (0–6 mm in size; pH 5.5–5.8; NPK complex 14–16–18 + microelements; with wetting agent; EC 2.5 mS·$cm^{-1}$). Once tomato plantlets were 21 days old, they were moved into a greenhouse under the following conditions: day/night temperature of 20–22 °C/15–17 °C; light/dark length–14 h/10 h; mean RH of 70%. At 42 days old, plants were transferred in 12 L plastic pots using the same substrate, and ten different treatments with MaEs were applied. In brief, six MaEs ($MgSO_4$, $KNO_3$, $K_2SO_4$, $Ca(NO_3)_2$, $KH_2PO_4$, and KCl) alone or in different combination ($MgSO_4$ + $KNO_3$ + $KH_2PO_4$; $K_2SO_4$ + $Ca(NO_3)_2$ + KCl; the mix of it; and one element each day) in a final concentration of 1 g/plant/day were used (Table 1). The proportion of the elements used in the mix was 1:1. An amount of 30 mL of each solution was applied every day for 21 days along with 500 mL to 1 L of water. Microelements ($Na_2MoO_4$, $Na_2[B_4O_5(OH)_4]\cdot8H_2O$, Cu, Mn, Zn, Fe) in a final concentration of 0.02 g/plant/week were applied foliar. Control plants were given water during the whole experiment. The experiment was conducted using block design with three replicates per treatment.

**Table 1.** Macronutrients treatment scheme.

| Code | Treatments | Final Concentration |
|------|------------|---------------------|
| V1 | $MgSO_4$ | 1 g/plant/day |
| V2 | $KNO_3$ | 1 g/plant/day |
| V3 | $K_2SO_4$ | 1 g/plant/day |
| V4 | $Ca(NO_3)_2$ | 1 g/plant/day |
| V5 | $KH_2PO_4$ | 1 g/plant/day |
| V6 | KCl | 1 g/plant/day |
| V7 | Mix 1 ($MgSO_4$–$KNO_3$–$KH_2PO_4$) | 1 g/plant/day |
| V8 | Mix 2 ($K_2SO_4$–$Ca(NO_3)_2$–KCl) | 1 g/plant/day |
| V9 | Mix 3 ($MgSO_4$–$KNO_3$–$K_2SO_4$–$Ca(NO_3)_2$–$KH_2PO_4$–KCl) | 1 g/plant/day |
| V10 | Cyclic daily treatment (one macroelement each day)(day 1–$MgSO_4$; day 2–$KNO_3$; day 3–$K_2SO_4$, day 4–$Ca(NO_3)_2$;day 5–$KH_2PO_4$; day 6–KCl) | 1 g/plant/day |
| V11 | $H_2O$ (Control) | 1 g/plant/day |

## 2.3. Electrical Conductivity Measurements of the Nutrients

The flow of nutrients passing through the tomato stem was determined by measuring the intensity of the electrical current generated by each species of nutrient. The measurements of the electrical signals were performed using a set of stainless-steel electrodes due to their biocompatible proprieties but also to avoid electrochemical reaction between electrodes and ions. This current was measured as the sum of the current generated by the nutrients and the current created by applying a potential difference of 9 V DC. To eliminate the electrical noises generated by internal and external factors into the electrical circuit used for measurements, the electrical circuit was grounded by connecting to the standard grounding socket of the power supply system.

The electric current was measured when plants were 42 days old for long periods of time (24 h), for 59 days, starting daily at 10:00 AM. The measurements were repeated 3 times during 3 different plant phenological stages: growth stage 2 (BBCH 201–259), flowering (BBCH 601–605), and the development of fruit (BBCH 701–705). The electrodes punctured the stem of the tomato plant to intersect the xylem. The distance between the two electrodes was about 40 cm, and the lower electrode was placed at a height of 25 cm above the soil (Figure 1). For current monitoring and recordings, a Keysight B2981A Femto/Picoammeter (fA), Santa Rosa, CA, USA, was used. The device can measure currents in the range of 0.01 fA–20 mA.

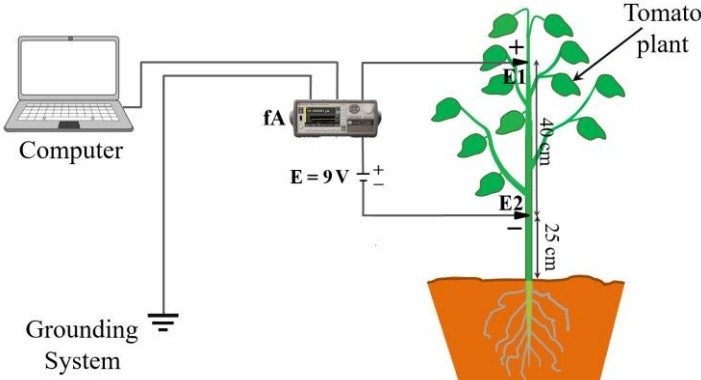

**Figure 1.** The electrical scheme used to monitor the electrical signal generated by the flow of nutrients through the xylem of tomato plants.

## 2.4. Leaf Gas Exchange Measurements

The photosynthetic rate ($\mu$mol $CO_2$ m$^{-2}$ s$^{-1}$) and the sub-stomatal $CO_2$ (Ci, vpm) were measured during the plant phenological stage BBCH 703–704 in the time interval 9–11 AM by using the Portable Photosynthetic System ADC Lci Bioscientific Ltd., Camas, WA, USA, with the leaf chamber area of 6.25 cm$^2$. The measurements were done in triplicate, with 10 reads for each determination.

## 2.5. Determination of Nutritional Composition of Tomato Fruits

The nutritional composition of tomato fruits (lipid, protein, fiber, dietary fiber, carbohydrate) was determined in triplicate. Samples were first oven dried at 105 °C until the weight became constant (AOAC, 2000). The lipid content was done by using the Soxhlet method and petroleum ether as solvent. The extraction was carried out for 8 h. % Lipid = Weight of lipid extracted × 100/Weight of dry sample (AOAC, 2000). The protein content was determined by using the Kjeldhal method and a conversion factor of 6.25 (AOAC, 2000). Fiber content was determined by the method described in AOAC, 2000. The dietary fiber was determined by enzymatic digestion of defatted and gelatinized samples [33]. The potentially available carbohydrate content was calculated by subtracting the total percentages of moisture, protein, lipid, ash, and fiber from 100%. Ash content was determined by incinerating 20 g of the samples at 550 °C for 8 h [34]. The energy was calculated according to the following formula:

Energy value (kcal) = Proteins (g) × 4 kcal/g + Carbohydrates (g) × 4 kcal/g + Lipids (g) × 9 kcal/g + Fibers (g) × 2 kcal/g.

## 2.6. Lycopene Content

The samples were first homogenized in a homogenizer. A 20 mL mixture of hexane–0.05% (*w/v*) butylated hydroxytoluene (BHT) in acetone–EtOH, 2:1:1 (*vol/vol*) was added in every flask containing 0.6 g of finely ground dry sample. The flasks were agitated continuously on a magnetic stirrer plate for 15 min on ice. After shaking, 3 mL of deionized water were added to each flask followed by another 5 min of agitation on ice. Then, the samples were left at room temperature for 5 min to allow the separation of polar and nonpolar layers. The absorbance of the hexane solution containing the lycopene was measured at 503 nm on a UV-VIS spectrophotometer T60 U PG Instruments Limited, UV WIN®version 5.05, Hertfordshire, UK, by using hexane as blank. The lycopene content was expressed as mg/100 g dry weight [35].

## 2.7. Determination of Antinutritional Factors in Tomato Fruits

### 2.7.1. Phytic Acid

An amount of 2 g of powdered sample was added in 100 mL 2% HCl and left to react for 3 h. Then, the solution was filtered through a double layer of harden filter paper and 50 mL of filtrate was added in 100 mL distilled water. To this solution was added an amount of 10 mL 0.3% ammonium thiocyanate as indicator and then was titrated with standard $FeCl_3$ solution (1.95 mg Fe/mL) until a brownish, persistent (5 min) coloration was obtained. The percentage of phytic acid was calculated using the formula: Phytic acid (mg/100 g) = Y × 1.19 × 100/Sample weight, where, Y = titer value × 1.95 mg [36].

### 2.7.2. Tannin

Two g of the sample was added into a beaker containing 50 mL of distilled water and heated to 60 °C. Then, the solution was filtered and the residue was discarded. A volume of 10 mL of 4% copper acetate solution was added to the hot filtrate and boiled again for 10 min. The precipitate was filtered and the filtrate was discarded. The residue was dried using filter paper, and the dried sample was weighted. Then, the samples were incinerated in a muffle furnace at 550 °C, cooled in a desiccator, and then reweighed. The difference between the weight of the sample before ashing and the ash residue after incineration represented the tannins content [37].

### 2.7.3. Oxalate

The total oxalic acid content of the powdered sample was determined by the method of Umogbai et al. [38], as follows: 2 g of finely ground sample was added in 190 mL of distilled water. Ten mL 6 M HCl solution was added to each sample, and the mixture was digested for 1 h at 100 °C. Then, the samples were cooled, made up to 250 mL, and then filtered. Four drops of methyl red indicator were added in each sample, which was followed by concentrated $NH_4OH$ until the solution turned faint yellow. Then, the solution was heated to 100 °C, allowed to cool, and filtered to remove precipitate containing ferrous irons. Then, the filtrates were heated to 90 °C and 10 mL of 5% $CaCl_2$ solution was added with constant stirring. After cooling, the samples were left overnight. Then, the mixtures were filtered through Whatman No. 4 filter paper. Then, the precipitates were washed several times with distilled water and dissolved in 5 mL 25% $H_2SO_4$. The resultant solution was maintained at 80 °C and titrated against 0.5% $KMnO_4$ until the pink color persisted for approximately one minute. A blank was run for the test sample [38]. The oxalate content of samples was related to the volume of $KMnO_4$ used for titration where 1 mL $KMnO_4$ = 2.24 mg oxalate [39].

### 2.7.4. Saponin

The saponin contents of the samples were determined following the AOAC (1990) method. Briefly, 2 g of sample was folded in a filter paper and extracted by refluxing in a Soxhlet extractor. Extraction was first done with 200 cm$^3$ of acetone for 3 h and then with 100 cm$^3$ MeOH for another 3 h. After the second extraction, the samples were oven-dried, left to cool at room temperature, and then weighed. Saponin content was calculated by using the following formula: Saponin mg/100 g = A − B × 100/Sm [40], where A = mass of flask and extract; B = mass of empty flask; Sm = sample mass [41].

### 2.7.5. Alpha–Amylase Inhibitors

A total of 250 μL of extract was placed in a tube and 250 μL of 0.02 mL sodium phosphate buffer, pH 6.9, containing α-amylase solution was added. This solution was pre-incubated at 25 °C for 10 min, after which 250 μL of 1% starch solution in 0.02 M sodium phosphate buffer pH 6.9 was added at a timed interval and then further incubated at 25 °C for 10 min. The reaction was stopped by adding 500 μL of dinitrosalicylic acid (DNS) reagent. Then, the tubes were incubated at 100 °C for 5 min and cooled to room temperature. The reaction mixture was diluted with 5 mL of distilled water, and the absorbance was measured at 540 nm [42]. A control was prepared using the same procedure replacing the extract with distilled water. The α-amylase inhibitory activity was calculated as % Inhibition = [(A control − A samples)/A control] × 100, where A control = absorbance of each control and A samples = the net absorbance of each sample. The net absorbance of each sample was calculated using the following equation: A samples = A test − A blank, where A test = absorbance of each test and A blank = absorbance of each blank. Concentrations of extract resulting in 50% inhibition of enzyme activity (IC50) were determined [43].

### 2.7.6. Trypsin Inhibitors

One g of finely ground sample was added in 50 mL of 0.5 M NaCl solution. The mixture was stirred for 30 min at room temperature and then centrifuged at 10,000× $g$ rpm in a Beckman JA-20 rotor, Beckman Coulter, Indianapolis, IN, USA. The supernatant was filtered through a Whatman filter paper (Whatman No.1), Merck, Darmstadt, Germany. The filtrate (extract volume) was used for the assay. To 2 mL of trypsin standard solution, 1 mL of trypsin inhibitor (1 mg/mL of trypsin in 0.1 M HCl) was added and incubated for 10 min at 37 °C for 10 min. A blank of 5 mL substrate (1% casein substrate in 0.1 M phosphate buffer pH 7.7) was prepared in a test tube (with no trypsin inhibitor extract added). The content in the test tube was left to rest for 10 min, and the reaction was stopped by adding 3 mL of 5% TCA. Then, the solution was filtered and absorbance was measured spectrophotometrically at 410 nm. The trypsin inhibitor activity was expressed as the number of trypsin unit inhibited (TUI) per unit weight of the sample analyzed: TUI/mg = (B − A)/0.1 × F, where B = absorbance of test sample solution, A = absorbance of the blank, F = (1 × Vf/Va × D)/W (W = weight of sample, Vf = total volume of extract used in the assay, D = dilution factor, Va = volume of standard) [44].

### 2.8. Minerals Bioavailability

Minerals (Ca, Fe, Zn) were determined by atomic absorption spectrometry [45]. The minerals bioavailability was predicted by calculating the molar ratios of antinutrient (e.g., phytic acid and oxalate)/mineral. The critical values used to predict a favorable bioavailability of minerals were as follows: phytate [Phy]/zinc [Zn] < 15; calcium [Ca]/phytate [Phy] < 6; phytate [Phy]/iron [Fe] < 1; phytate × calcium [Phy × Ca]/zinc [Zn] < 0.5; oxalate [OX]/calcium [Ca] < 2.5 [46].

### 2.9. Statistical Analysis

The data are expressed as the means ± standard deviation (SD). One-way analysis of variance (ANOVA) was used to see the influence of the treatments on the nutritional and antinutrient composition

of tomato fruits cv. Brillante F1. The significant differences between treatments were established by using Tukey's post hoc test with a degree of confidence of 95% ($p < 0.05$).

## 3. Results and Discussion

### 3.1. Electrical Signals of the Nutrients

To identify the shape of the electrical signal specific for each MaE used in the experiment ($MgSO_4$, $KNO_3$, $K_2SO_4$, $Ca(NO_3)_2$, $KH_2PO_4$, and $KCl$), the electric current was measured during a 24 h time period. The recorded results showed that the signals differed depending on the nutrient supplied to the plant (Figure 2). The $Ca(NO_3)_2$ solution generated the highest signal in the plants, with a double value compared to the signals measured for the other nutrients. In contrast, the $KH_2PO_4$ solution generated the smallest signal, but nevertheless, it was higher than that recorded for the control plants (V11), which received only water. However, several common elements have been observed among the all signals. For instance, immediately after the application of the treatments (after 10:30 AM), the signal values increased, stayed high during the day, and then sharply increased at the end of the day (18:00–20:00 h), after which they decreased and remained low during the night. In the morning, after the sunrise (06:30 AM), the signal values started to increase again (Figure 2).

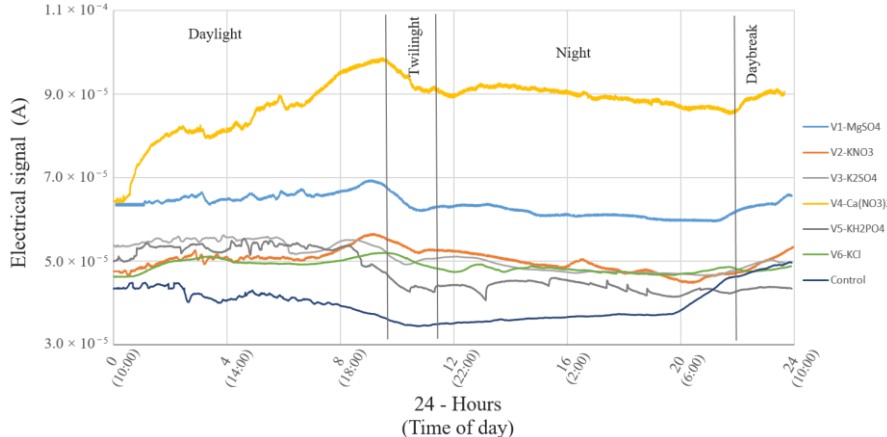

**Figure 2.** The unique daily imprint of the electrical signal generated by $MgSO_4$ (V$_1$), $KNO_3$ (V$_2$), $K_2SO_4$ (V$_3$), $Ca(NO_3)_2$ (V$_4$), $KH_2PO_4$ (V$_5$), $KCl$ (V$_6$), and $H_2O$ (V11—control) in the tomato plants during 24 h.

This trend indicates that the plant nutrition is closely correlated with the photosynthesis process, when an ionic depression is created at the root system level. Therefore, it can be observed that the shape of the signals varies during the day in accordance with the period in which photosynthesis takes place, which means that it depends on the physiological processes of the plant and can be correlated with the circadian rhythm (Figure 2).

The shape of the signal measured for each nutrient is necessary in order to perform the deconvolution of the unique signal recorded in the basic signals. The use of a mathematical algorithm that identifies the basic signals corresponding to each nutrient can be the key to design a biosensor that can be used in the technological process.

### 3.2. Correlating the Electrical Signals with the Leaf Gas Exchange Processes

To correlate the electric signals recorded for every MaEs or mixture of MaEs with the circadian rhythm of the biochemical and physiological processes dependent on the nutrient consumption, we measured some photosynthetic parameters (photosynthesis rate and sub-stomatal $CO_2$). These measurements were done in order to understand the shape of the signal before the end of the day.

The results clearly suggested that when the light intensity decreased during the twilight, the photosynthesis process was reduced (Figure 3b) and dominated by the respiration. During this period, the registered amount of $CO_2$ released was increased (Figure 3c). For instance, the amount of $CO_2$ recorded for V7, also, for the rest of the treatments, showed a maximum at 19.00 h (Figure 3c).

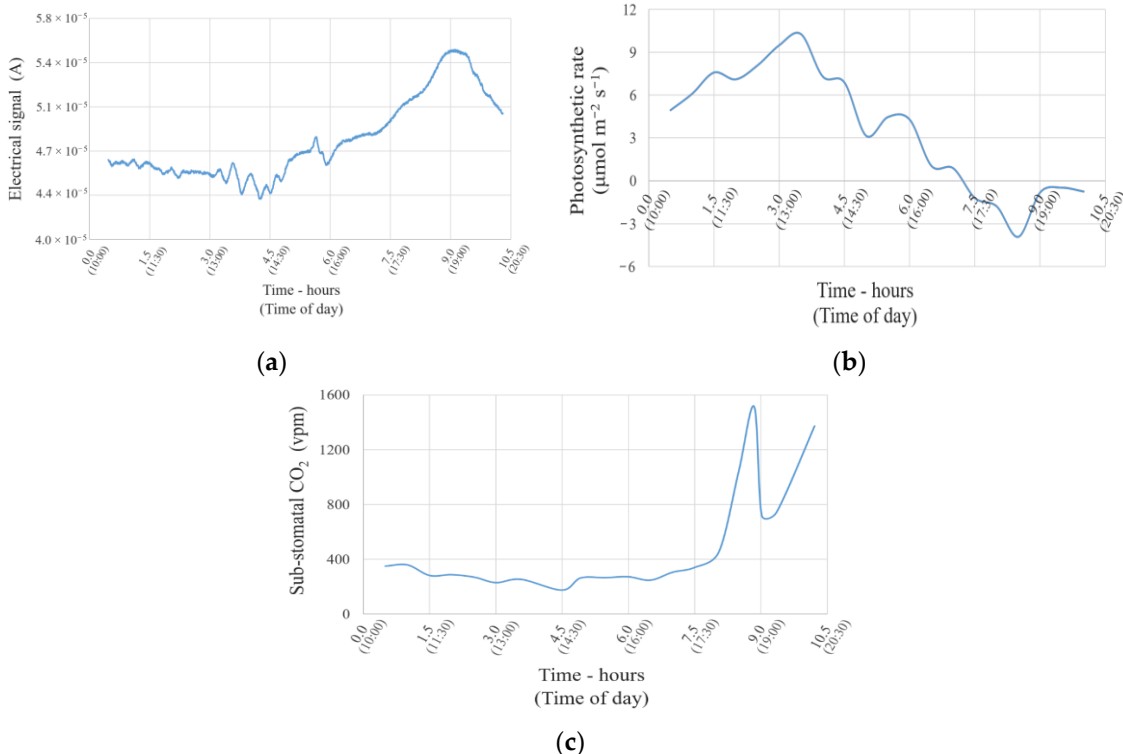

**Figure 3.** The electrical signal (**a**), the photosynthesis rate (**b**), and the amount of $CO_2$ (**c**) registered for V7 ($MgSO_4$– $KNO_3$– $KH_2PO_4$).

Therefore, the measurements recorded for the photosynthetic parameters showed that the peak obtained for the electrical signal at the end of the day (Figure 3a) was due to the 5-fold increase of the amount of released $CO_2$ (Figure 3c) and the decrease in the rate of photosynthesis (Figure 3b).

*3.3. Analysis of the Electrical Signals during Different Plant Phenological Stages*

In order to develop a technological concept for a smart fertilization by using a biosensor based on the electrical signals, information about the electrical signal variation during different plant phenological stages is important.

Therefore, three phenological stages were chosen: growth stage 2 (BBCH 201–259)—R1, flowering (BBCH 601–605)—R2, and development of fruit (BBCH 701–705)—R3.

The results showed that for the Brillante F1 tomato cultivar, each phenological stage and treatment generate a different electrical signal. Five trends of electric signals that differ from one phenological stage to another have been identified:

- The electrical signal values decrease from one phenological stage to another, trend registered for V1—$MgSO_4$ (Figure 4a) and V3—$K_2SO_4$ (Figure 4c).
- The electrical signal values increase from one phenological stage to another, trend registered for V2—$KNO_3$ (Figure 4b), V6—$KCl$ (Figure 4f), V7—$MgSO_4$–$KNO_3$ –$KH_2PO_4$ (Figure 4g), and V8—$K_2SO_4$–$Ca(NO_3)_2$–$KCl$ (Figure 4h).
- The electrical signal values decrease and then increase from one phenological stage to another (V4—$Ca(NO_3)_2$) (Figure 4d).

- The electrical signal values increase and then decrease from one phenological stage to another, trends registered for V5—$KH_2PO_4$ (Figure 4e), V9—$MgSO_4 – KNO_3 – K_2SO_4 –Ca(NO_3)_2 – KH_2PO_4 – KCl$ (Figure 4i), and V10—cyclic daily treatment (Figure 4j).
- The electrical signal values overlap during the phenological stages taken into consideration (V11—$H_2O$) (Figure 4k).

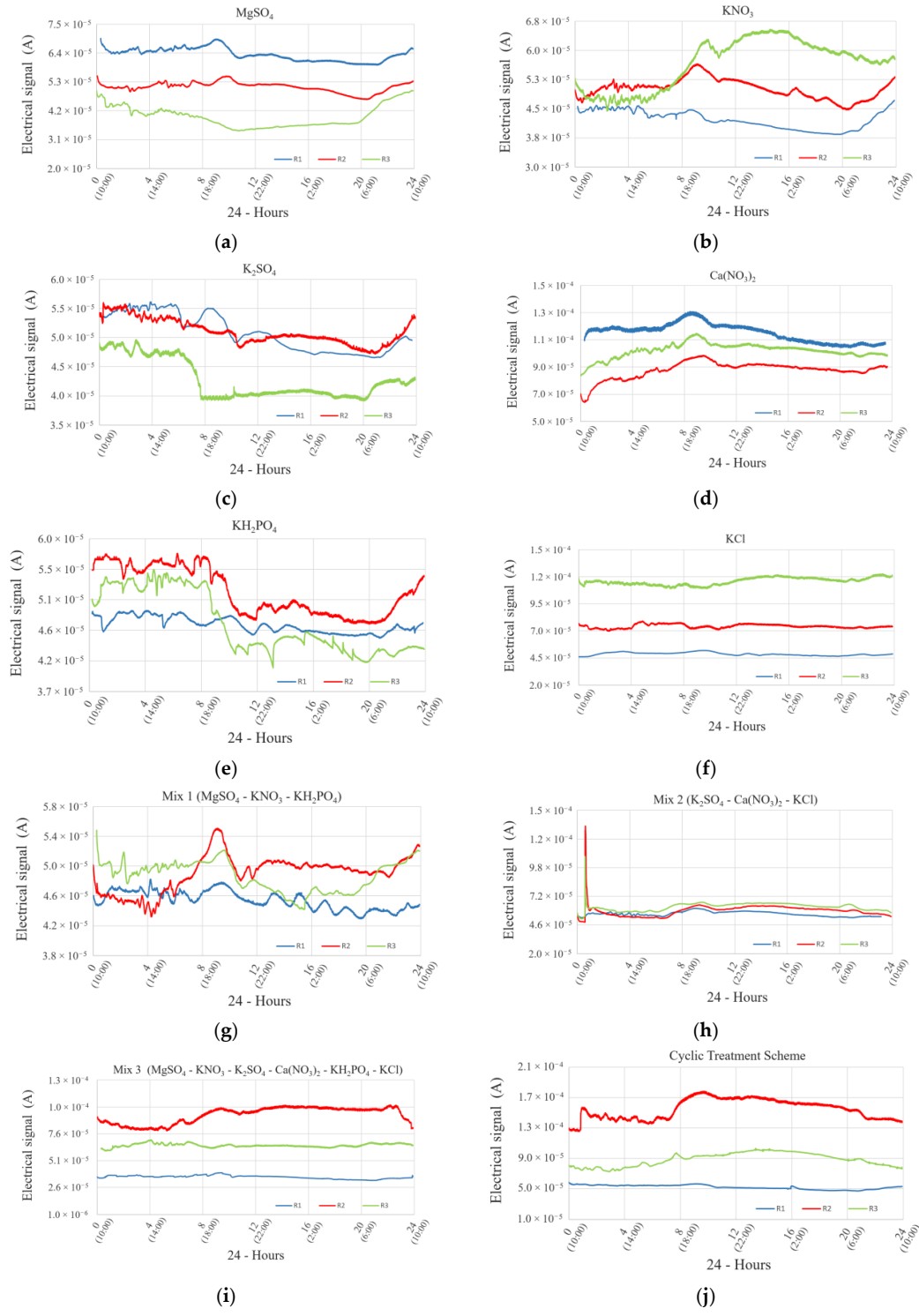

**Figure 4.** *Cont.*

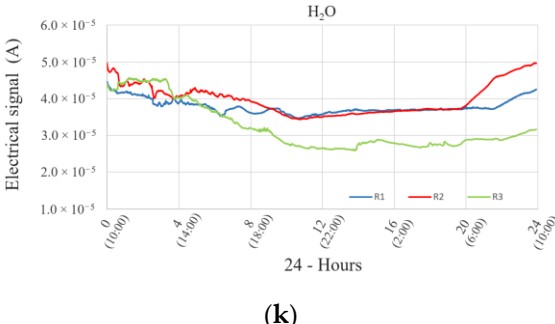

(**k**)

**Figure 4.** The electrical signals measured for different types of treatments during three phenological stages (R1, R2, and R3): (**a**) V1—MgSO$_4$; (**b**) V2—KNO$_3$; (**c**) V3—K$_2$SO$_4$; (**d**) V4—Ca(NO$_3$)$_2$; (**e**) V5—KH$_2$PO$_4$; (**f**) V6—KCl; (**g**) V7—Mix of MgSO$_4$–KNO$_3$–KH$_2$PO$_4$; (**h**) V8—Mix of K$_2$SO$_4$–Ca(NO$_3$)$_2$–KCl; (**i**) V9—Mix of MgSO$_4$–KNO$_3$–K$_2$SO$_4$–Ca(NO$_3$)$_2$–KH$_2$PO$_4$–KCl; (**j**) V10—cyclic treatment scheme and (**k**) V11—H$_2$O.

### 3.4. Influence of the Treatments on the Nutritional Composition of Tomato Fruits

The analysis on the nutritional composition of tomato fruits was done in order to understand the effect of the treatments on the quality of tomatoes. The results are shown in Table 2. The treatment that gave the best records for the protein, fiber, dietary fiber, total carbohydrates contents, and energy was the one with K$_2$SO$_4$ (V3), followed by that with KCl (V6). The data recorded for these treatments were significantly higher as compared to those registered for V9 treatment, which consisted in a standard mix of MaEs, which is usually used in the agricultural practices [47]. For instance, the protein content of the tomato fruits of the plants treated with K$_2$SO$_4$ was increased by 18.5%, the fiber was increased by 24.7%, the dietary fiber was increased by 16.1%, the carbohydrates were increased by 34.7% and the energy was increased by 24.1% compared to those derived from tomato plants treated with V9. Regarding the ash content, it was observed that for K$_2$SO$_4$ (V3) and KCl (V6) treatments, the values registered were among the lowest (0.58%—V3; 0.61%—V6), while for V9 treatment, it was the highest (0.74%).

**Table 2.** Nutritional values of tomato fruits cv. Brillante F1 treated with different major elements (MaEs).

| Treatment | Ash (%) | Lipid (g/100 g) | Protein (g/100 g) | Fiber (g/100 g) | Dietary Fiber (g/100 g) | Total Carbohydrates (g/100 g) | Energy (kcal/100 g) |
|---|---|---|---|---|---|---|---|
| V1 | 0.58 ± 0.009 f | 0.18 ± 0.000 ns | 2.53 ± 0.006 c | 1.03 ± 0.000 b | 0.61 ± 0.006 c | 3.06 ± 0.016 c | 26.05 ± 0.040 c |
| V2 | 0.68 ± 0.001 c | 0.18 ± 0.000 ns | 2.20 ± 0.000 j | 0.89 ± 0.003 g | 0.54 ± 0.000 g | 2.45 ± 0.002 j | 22.03 ± 0.005 j |
| V3 | 0.58 ± 0.002 f | 0.18 ± 0.000 ns | 2.63 ± 0.000 a | 1.06 ± 0.001 a | 0.65 ± 0.003 ab | 3.22 ± 0.001 a | 27.12 ± 0.008 a |
| V4 | 0.64 ± 0.001 d | 0.18 ± 0.000 ns | 2.37 ± 0.002 f | 0.95 ± 0.000 d | 0.60 ± 0.009 de | 2.75 ± 0.002 f | 23.99 ± 0.003 f |
| V5 | 0.69 ± 0.001 bc | 0.18 ± 0.000 ns | 2.23 ± 0.001 h | 0.89 ± 0.002 g | 0.57 ± 0.000 f | 2.49 ± 0.001 i | 22.29 ± 0.002 i |
| V6 | 0.61 ± 0.17 e | 0.18 ± 0.000 ns | 2.59 ± 0.001 b | 1.03 ± 0.000 b | 0.66 ± 0.001 a | 3.09 ± 0.018 b | 26.37 ± 0.071 b |
| V7 | 0.70 ± 0.001 b | 0.18 ± 0.000 ns | 2.32 ± 0.001 g | 0.92 ± 0.000 e | 0.59 ± 0.000 e | 2.56 ± 0.002 h | 22.95 ± 0.005 h |
| V8 | 0.65 ± 0.001 d | 0.18 ± 0.000 ns | 2.52 ± 0.000 d | 0.97 ± 0.005 c | 0.64 ± 0.001 b | 2.94 ± 0.003 d | 25.38 ± 0.004 d |
| V9 | 0.74 ± 0.001 a | 0.18 ± 0.000 ns | 2.22 ± 0.001 i | 0.85 ± 0.001 h | 0.56 ± 0.000 f | 2.39 ± 0.001 k | 21.71 ± 0.004 k |
| V10 | 0.70 ± 0.001 bc | 0.18 ± 0.000 ns | 2.37 ± 0.000 f | 0.90 ± 0.001 f | 0.60 ± 0.001 cd | 2.66 ± 0.002 g | 23.53 ± 0.010 g |
| V11 | 0.66 ± 0.000 d | 0.18 ± 0.000 ns | 2.51 ± 0.000 e | 0.95 ± 0.001 d | 0.64 ± 0.001 b | 2.91 ± 0.001 e | 25.21 ± 0.003 e |

Note: V1 = MgSO$_4$, V2 = KNO$_3$, V3 = K$_2$SO$_4$, V4 = Ca(NO$_3$)$_2$, V5 = KH$_2$PO$_4$, V6 = KCl, V7 = MgSO$_4$–KNO$_3$–KH$_2$PO$_4$, V8 = K$_2$SO$_4$–Ca(NO$_3$)$_2$–KCl, V9 = MgSO$_4$–KNO$_3$–K$_2$SO$_4$–Ca(NO$_3$)$_2$–KH$_2$PO$_4$–KCl, V10 = one macroelement each day, V11 = H$_2$O (Control). Different letters mean significant differences between the values according to Tukey's post hoc test ($p < 0.05$), ns = not significant.

The analysis of the electrical signals trend recorded during the development of fruit stage (BBCH 701–705) showed that for V3 treatment, it was registered to have the lowest electrical signal as compared with growth stage 2 (BBCH 201–259) and flowering stage (BBCH 601–605); for V6, it recorded the highest electrical signal as compared with the other two stages, while for the mix usually used in agriculture (V9), the electrical signal was higher than that for growth stage 2 (BBCH 201–259) but lower than that of the flowering stage. The values of the electrical signals registered for V3 varied

between $3.91 \times 10^{-5}$ A and $4.96 \times 10^{-5}$ A, for V6, they varied between $1.09 \times 10^{-4}$ A and $1.23 \times 10^{-4}$ A, and for V9, they varied between $6.00 \times 10^{-5}$ A and $7.04 \times 10^{-5}$ A (Figure 4).

These results suggest that for a high nutritional value of the fruits, the tomato plants from Brillante F1 cultivar need during the fruits formation and development a fertilizer based only on potassium. This finding is supported by Fontes et al. [48], who stated that during the fruit growth stage of tomato plants, K is the most absorbed nutrient. In addition, potassium is the key element for the production of quality tomato fruits [49]. In general, potassium is the nutrient known to improve the quality of the plants, being responsible for protein synthesis, enzyme activation, nutrient balance, carbohydrate metabolism, water movement, and many other processes [50]. Yang et al. [51] showed in an experiment on maize that the grains treated with NPK fertilizer had a higher protein content and grain quality compared to the grains treated with only NP fertilizer. Increases on the crop qualities due to the potassium application were also registered for beet, regarding the dry matter, the soluble, and refinable sugars [52] or for tomato in terms of dry matter, total soluble solids, vitamin C content, and firmness [53]. The improvement of fruit quality depends also on the potassium fertilizer forms. Therefore, studies have demonstrated that $K_2SO_4$ give a better or at least the same quality as KCl, but both are more effective than $KNO_3$ [54]. This trend was also observed in the fruit quality of Brillante F1 cultivar (Table 2).

### 3.5. Influence of the Treatments on the Lycopene Content of Tomato Fruits

Lycopene is the main pigment responsible for the red color of some fruits and vegetables [35,55,56]. In addition, it is known to have an important role in the prevention of different diseases [57]. The major sources of lycopene for human diet are represented by tomato fruits. The lycopene content in ripe tomato can reach 83%, but the value can vary depending on many factors such as genotype, maturity, environmental conditions (temperature and light), or cropping systems (mineral nutrition, irrigation, grafting) [55,56,58]. Information about the effect of mineral nutrition on the lycopene content of tomatoes is limited and sometimes contradictory [59,60]. In addition, most of the studies focused on the influence of just one or maximum three MaEs of different concentrations on the lycopene content of the plants. For instance, Serio et al. [58], Trudel and Ozbun [61] or Oded and Uzi [62] reported increases of the lycopene content of tomatoes at increased concentrations of potassium, while Fontes et al. [63] concluded that lycopene do not depend on the potassium level. Saito and Kano [64] have found a positive correlation between the increased levels of phosphorus and the lycopene content of tomatoes, while Zdravković et al. [65] recorded no effect of phosphorus fertilizers on the lycopene content. Moreover, Oke et al. [66] and Adebooye et al. [67] have reported that high P amounts decreased the lycopene content of tomato fruits. Dumas et al. [60] reported that lycopene content increases in the context of the increased amount of nitrogen fertilizer applied. The same observation was done by Paiva et al. [68] regarding the calcium fertilizers. To our knowledge, no reports have yet been published regarding a comparison between the effect of six different MaEs commonly used for fertilization and their combination on the lycopene content of tomatoes. The results of the treatments' influence on the lycopene content of tomato fruits cv. Brillante F1 are shown in Figure 5. As in the case of nutritional composition, the tomato fruits of the plants treated with $K_2SO_4$ (V3) had the highest lycopene content (9.59 mg/100 g). As mentioned before, the electrical signals' trends recorded for this treatment during the development of fruit stage (BBCH 701–705) were the lowest as compared with growth stage 2 (BBCH 201–259) and the flowering stage (BBCH 601–605). The value of the electrical signal varied between $3.91 \times 10^{-5}$ A and $4.96 \times 10^{-5}$ A (Figure 4c). No significant differences were registered as compared to V2 treatment, which consisted in $KNO_3$ (9.33 mg/100 g), or to V8, which was a mix of three MaEs ($K_2SO_4$, $Ca(NO_3)_2$ and KCl)—9.40 mg/100 g. For both of these treatments, the electrical signals' trend recorded were the highest during the fruit development stage (BBCH 701–705), as compared with growth stage 2 (BBCH 201–259) and flowering stage (BBCH 601–605) (Figure 4b,h).

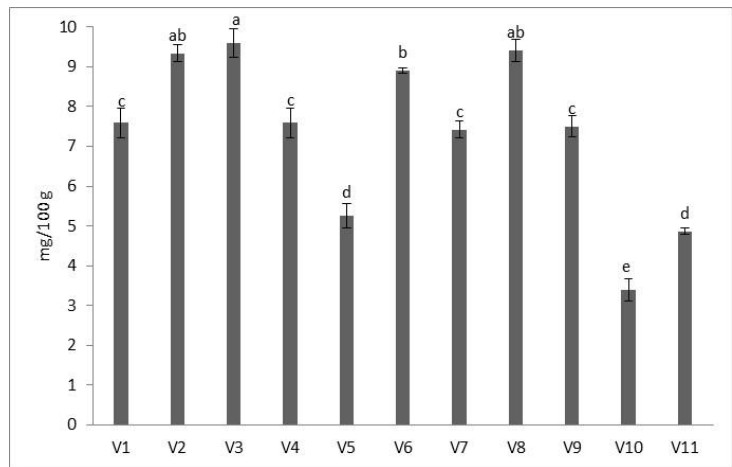

**Figure 5.** Lycopene content of tomato fruits cv. Brillante F1 treated with different fertilizers. Data are means ± SD. V1 = $MgSO_4$, V2 = $KNO_3$, V3 = $K_2SO_4$, V4 = $Ca(NO_3)_2$, V5 = $KH_2PO_4$, V6 = KCl, V7 = $MgSO_4$–$KNO_3$–$KH_2PO_4$, V8 = $K_2SO_4$–$Ca(NO_3)_2$–KCl, V9 = $MgSO_4$–$KNO_3$–$K_2SO_4$–$Ca(NO_3)_2$–$KH_2PO_4$–KCl, V10 = one macroelement each day, V11 = $H_2O$ (Control). Different letters represents significant differences between the values according to Tukey post hoc test ($p < 0.05$).

The lowest lycopene content was recorded for the plants treated every day with a different MaE (V10—3.40 mg/100 g). The plants treated with the standard mix of MaEs (V9) had a lycopene content of the tomato fruits of 28% lower than those treated with $K_2SO_4$ (V3) or of 24% lower as compared with those treated with $KNO_3$. The electrical signals recorded for both V9 and V10 treatments showed that during the fruit development stage (BBCH 701–705), the amplitude of the signals was higher than that for growth stage 2 (BBCH 201–259) but lower than that of the flowering stage (Figure 4i,j). As in the case of the nutritional composition, it was observed that different potassium fertilizer forms had different effects on the lycopene content of tomato fruits. Therefore, the $K_2SO_4$ fertilizer gave the highest content of lycopene, followed by $KNO_3$ fertilizer, between which the differences were not significantly higher, and by KCl fertilizer for which the lycopene content was significantly lower than that of V3 treatment, but not significantly higher than that of V2 treatment. The analysis of the electrical signals' trends showed that for $K_2SO_4$ fertilizer, the registered value varied between $3.91 \times 10^{-5}$ A and $4.96 \times 10^{-5}$ A, for $KNO_3$, it varied between $4.43 \times 10^{-5}$ A and $6.53 \times 10^{-5}$ A, and for KCl, it varied between $1.09 \times 10^{-4}$ A and $1.23 \times 10^{-4}$ A (Figure 4c).

In addition, it was interesting to observe that for lycopene, as for the nutritional composition, the highest values were registered for treatments whose electrical signals during the fruit development stage were the lowest or the highest, while the lowest values were those for which the amplitude of the signals was higher than that of growth stage 2 (BBCH 201–259) but lower than that of the flowering stage.

In addition, it was observed that among the treatments where only one MaE was applied at once, those based on potassium ($K_2SO_4$, $KNO_3$, KCl) gave the best records for the lycopene content, which was followed by those based on magnesium ($MgSO_4$) and calcium ($Ca(NO_3)_2$) and finally by a phosphorus-based fertilizer ($KH_2PO_4$). The high values of lycopene content registered for K fertilizers might be related to the important role of this element in the process of carotenoid biosynthesis. Potassium is known to be involved in the formation of enzymes (e.g., acetic thiokinase) implied in the synthesis of the first precursor of carotenoids—isopentenyl diphosphate (IPP) or in the regulation of pyruvate and glyceraldehyde 3–phosphate, known as percursors of IPP [61,69]. Our results are in accordance with those reported by Fanasca et al. [69] who in a study regarding the effect of K, Mg, and Ca on the tomato fruit quality have found that the highest lycopene content was registered on plants grown in high concentrations of K, followed by Mg and Ca. Taking into account that the potassium fertilization regardless of the forms used gave the best nutritional values and lycopene content, it can

be assumed that during the fruit growth stage, the tomato plants cv. Brillante F1 do not need all the MaEs at once, but they do need mainly potassium fertilizers in order to get high-quality fruits.

### 3.6. Influence of the Treatments on the Antinutrient Composition of Tomato Fruits

The presence of antinutritional components in food such as phytic acid, tannins, oxalates, saponins, $\alpha$-amylase, or trypsin inhibitors is known to affect the human nutrition by influencing the bioavailability of different minerals (Fe, Ca, and Zn) or enzymes ($\alpha$-amylase or trypsin), making them unavailable for intestinal absorption [70,71]. The antinutrient composition of tomato fruits cv. Brillante F1 under the influence of different MaEs treatments is shown in Table 3. The results varied depending on the nutrient treatment and the component analyzed.

**Table 3.** Antinutrient composition of tomato fruits cv. Brillante F1 treated with different MaEs. Data are means ± SD.

| Treatment | Phytic Acid (mg/100 g) | Tannin (g/100 g) | Oxalate (g/100 g) | Saponin (g/100 g) | $\alpha$-Amylase Inhibitor IC50 (mg/mL) | Trypsin Inhibitor (TUI/mg) |
|---|---|---|---|---|---|---|
| V1 | 0.23 ± 0.00 cd | 0.33 ± 0.00 c | 0.16 ± 0.00 e | 0.18 ± 0.00 d | 7.51 ± 0.10 c | 12.35 ± 0.04 e |
| V2 | 0.20 ± 0.00 g | 0.28 ± 0.00 f | 0.14 ± 0.00 h | 0.16 ± 0.00 g | 7.67 ± 0.01 b | 12.50 ± 0.10 d |
| V3 | 0.24 ± 0.00 b | 0.34 ± 0.00 ab | 0.17 ± 0.00 c | 0.19 ± 0.00 ab | 7.68 ± 0.01 ab | 12.63 ± 0.03 c |
| V4 | 0.22 ± 0.00 e | 0.31 ± 0.00 d | 0.16 ± 0.00 f | 0.18 ± 0.00 d | 7.72 ± 0.01 ab | 12.71 ± 0.01 bc |
| V5 | 0.21 ± 0.00 f | 0.29 ± 0.00 f | 0.15 ± 0.00 g | 0.17 ± 0.00 f | 7.72 ± 0.00 ab | 12.72 ± 0.00 abc |
| V6 | 0.25 ± 0.00 a | 0.34 ± 0.00 b | 0.18 ± 0.00 c | 0.19 ± 0.00 ab | 7.72 ± 0.00 ab | 12.73 ± 0.01 abc |
| V7 | 0.23 ± 0.00 d | 0.30 ± 0.00 e | 0.16 ± 0.00 d | 0.17 ± 0.00 de | 7.73 ± 0.01 ab | 12.73 ± 0.01 abc |
| V8 | 0.25 ± 0.00 a | 0.33 ± 0.00 c | 0.18 ± 0.00 b | 0.19 ± 0.00 b | 7.74 ± 0.01 ab | 12.74 ± 0.01 ab |
| V9 | 0.22 ± 0.00 e | 0.29 ± 0.00 f | 0.17 ± 0.00 d | 0.17 ± 0.00 ef | 7.74 ± 0.00 ab | 12.74 ± 0.00 ab |
| V10 | 0.23 ± 0.00 c | 0.32 ± 0.00 c | 0.18 ± 0.00 b | 0.19 ± 0.00 c | 7.74 ± 0.00 ab | 12.76 ± 0.02 ab |
| V11 | 0.25 ± 0.00 a | 0.34 ± 0.00 a | 0.20 ± 0.00 a | 0.20 ± 0.00 a | 7.76 ± 0.01 a | 12.77 ± 0.00 a |

Note: V1 = $MgSO_4$, V2 = $KNO_3$, V3 = $K_2SO_4$, V4 = $Ca(NO_3)_2$, V5 = $KH_2PO_4$, V6 = KCl, V7 = $MgSO_4$–$KNO_3$–$KH_2PO_4$, V8 = $K_2SO_4$–$Ca(NO_3)_2$–KCl, V9 = $MgSO_4$–$KNO_3$–$K_2SO_4$–$Ca(NO_3)_2$–$KH_2PO_4$–KCl, V10 = one macroelement each day, V11 = $H_2O$ (Control). Different letters mean significant differences between the values according to Tukey posthoc test ($p < 0.05$). IC50 = half maximal inhibitory concentration; TUI = trypsin units inhibited.

Phytic acid, apart from its important roles in the storage of phosphorus, mRNA export, or chromatin remodeling, is known to bind different essential minerals such as Fe, Zn, or Ca and to reduce their availability to human nutrition. In addition, it was demonstrated that the presence of $K^+$, $Ca^{2+}$, and $Zn^{2+}$ cations in substrate can enhance the synthesis of phytic acid [72]. In our experiment, the phytic acid concentration in tomato fruits varied between 0.20 mg/100 g (V2) and 0.25 mg/100 g (V6, V8). Both of the values were registered for tomato fruits from plants treated with potassium fertilizers. This suggested that potassium can influence the concentration of phytic acid in tomato fruits, depending on its type. When plants were treated with $KNO_3$ (V2), the concentration of phytic acid registered in the fruits was the lowest (0.2 mg/100 g), while when the treatment consisted in KCl (V6) or the mix composed of $K_2SO_4$ + $Ca(NO_3)_2$ + KCl (V8), the concentration was maximum (0.25 mg/100 g). The electrical signals recorded for all these treatments during the development of the fruit stage (BBCH 701—705) were the highest as compared to growth stage 2 (BBCH 201–259) and flowering stage (BBCH 601–605) (Figure 4).

Comparing the results registered for potassium fertilizers with those for the mix usually used in agriculture (V9), it was observed that the phytic acid concentration of the fruits from the plants treated with V9 was significantly higher than that of the tomato plants treated with V2 and significantly lower than that of the plants treated with V8. Taking into account the phytic acid concentration, the bioavailability of zinc, calcium, and iron in the tomato fruits of cv. Brillante F1 were calculated (Table 4). The [Phy]/[Zn] molar ratio varied between 6.20 and 9.63 mg/100 g, being less than the critical value (<15 favorable for zinc absorption). According to the WHO, a molar ratio greater or equal to 15 indicates a low zinc bioavailability (10–15%), between 5 and 15 indicates a moderate bioavailability (30–35%), and below 5 indicates a high zinc bioavailability (50–55%) [70]. The tomato fruits from

cv. Brillante F1, regardless of the treatment applied, had moderate zinc availability for the human nutrition. The best value, 6.20 mg/100 g FW was registered for the fruits of the plants treated every day with a different MaE (V10), whose electrical signal during the fruit development stage was higher than that for growth stage 2 (BBCH 201–259) but lower than that of the flowering stage and value varied between $7.26 \times 10^{-5}$ A and $1.03 \times 10^{-4}$ A (Figure 4). The lowest zinc availability as compared to that recorded for V10 was seen for the fruits of the plants treated with $MgSO_4$ (V1)—9.63 mg/100 g. For this, the electrical signal value during the fruit development stage was the lowest as compared with the other two stages taken into consideration and varied between $3.43 \times 10^{-5}$ A and $4.97 \times 10^{-5}$ A (Figure 4). Zinc bioavailability from the [Phy]/[Zn] complex can also be affected by calcium ions, which interact with zinc ions, creating a synergism between them, resulting in the Ca:Zn:Phy complex with low solubility [70]. In our study, the [Phy × Ca]/[Zn] molar ratio was below the critical value (>0.5 low zinc availability), suggesting that calcium does not affect the zinc availability (Table 4). An optimum zinc availability for human is important for a proper functioning of various systems as epidermal, gastrointestinal, central nervous, immune, skeletal, and reproductive [73].

**Table 4.** Bioavailability of zinc, calcium, and iron in the tomato fruits of cv. Brillante F1.

| Sample | [Phy]/[Zn] mg/100 g FW | [Ca]/[Phy] mg/100 g FW | [Phy]/[Fe] mg/100 g FW | [Phy×Ca]/[Zn] mol/kg FW | [OX]/[Ca] mg/100 g FW |
|---|---|---|---|---|---|
| V1 | 9.63 | 6.38 | 35,881.70 | 0.001 | 119.20 |
| V2 | 8.97 | 5.88 | 16,259.30 | 0.001 | 121.01 |
| V3 | 8.76 | 5.72 | 11,768.87 | 0.001 | 122.53 |
| V4 | 8.48 | 5.51 | 11,484.31 | 0.001 | 123.78 |
| V5 | 7.51 | 4.77 | 9124.49 | 0.001 | 124.66 |
| V6 | 6.26 | 3.84 | 9051.10 | 0.001 | 128.30 |
| V7 | 6.36 | 3.84 | 8991.27 | 0.001 | 134.29 |
| V8 | 6.32 | 3.79 | 8687.78 | 0.001 | 138.82 |
| V9 | 6.30 | 3.77 | 7778.62 | 0.001 | 142.27 |
| V10 | 6.20 | 3.70 | 6023.67 | 0.001 | 146.05 |
| V11 | 6.22 | 3.70 | 5732.63 | 0.001 | 148.80 |
| Critical values | <15 Favorable for zinc absorption | <6 Favorable for calcium absorption | >1 Low iron bioavailability | >0.5 Low zinc bioavailability | >2.5 Low calcium bioavailability |

Note: V1 = $MgSO_4$, V2 = $KNO_3$, V3 = $K_2SO_4$, V4 = $Ca(NO_3)_2$, V5 = $KH_2PO_4$, V6 = KCl, V7 = $MgSO_4$–$KNO_3$–$KH_2PO_4$, V8 = $K_2SO_4$–$Ca(NO_3)_2$–KCl, V9 = $MgSO_4$–$KNO_3$–$K_2SO_4$–$Ca(NO_3)_2$–$KH_2PO_4$–KCl, V10 = one macroelement each day, V11 = $H_2O$ (Control). The data represent the mean of three determinations per analyzed component. FW = fresh weight; OX = oxalates.

Calcium is another important mineral for human nutrition accounting for 1–2% of adult human body weight. Most of the calcium (99%) is found in bones and teeth. Poor intestinal absorption of calcium is one of the causes of osteoporosis or reduced bone mass [74]. Therefore, consuming foods with optimum calcium bioavailability is very important. Phytic acid is known to inhibit the calcium availability. In our study, the results recorded for [Ca]/[Phy] molar ratio (3.70–6.38 mg/100 g) suggested that except for the plants treated with $MgSO_4$ (V1), calcium was favorable for absorption (<6). As in the case of zinc, the best bioavailability was registered for the tomato fruits of the plants treated every day with a different MaEs (V10).

Apart from reducing the bioavailability of zinc and calcium, phytic acid is considered to be the main inhibitor of iron absorption. The lack of iron in human nutrition leads to different types of anemia. The World Health Organization estimates that two billion people are suffering from anemia. Of this, one billion have anemia because of iron deficiency caused mainly by low intestinal absorption. The molar ratio between phytic acid and iron is used as indicator of iron bioavailability [75]. In our study, the iron availability in the tomato fruits of cv. Brillante F1 was very low, the molar ratio

being bigger than the critical value—1. To be optimum, the ratio should be 1:1 or 0.4:1 in cereal or legume-based meals. Phytic acid is known to have a negative impact on iron absorption starting with a low concentration of 2–10 mg/meal, and it is also dose dependent [75].

Iron bioavailability is also reduced by tannins [46]. Tannins are a group of polyphenols with antinutritional properties known to affect the protein digestibility, the activities of amylase, lipase, and trypsin, or the iron absorption. A diet rich in tannins has a negative impact on the cellulose and intestinal digestion. The concentration of tannins in plants varies depending on specie [76]. For instance, in tomato, potato, eggplant, or pepper, the encountered amount of tannins can be of 0.19 mg/100 g, in legumes (lentils, chickpeas, beans, soybean) varies between 1.8 and 18 mg/g and in tubers (carrot, sweet potatoes), it can reach 4.18–6.72 mg/100 g [77]. In the tomato fruits of the cv. Brillante F1, the tannins concentration ranged, depending on the treatment, from 0.28 mg/100 g (V2–$KNO_3$) to 0.34 mg/100 g (V3–$K_2SO_4$, V6–KCl, V11–control). For the plants treated with $K_2SO_4$, it also registered the highest nutritional values (Table 2), the presence of the highest content of tannin being a drawback for the tomato quality (Table 4). The amount of tannins in tomato fruits of the plants treated with the standard mix of MaEs (V9) was not significantly different as compared with that recorded in the tomato fruits of the plants treated with $KNO_3$ (V2). It was observed that the potassium fertilizers depending on their forms, as in the case of nutritional value, can have different effects on the tannin content. For instance, the plants treated with $KNO_3$ had the lowest tannin content in the fruits. No significant difference was seen as compared with the plants treated with $KH_2PO_4$ (V5). On the other side, the plants treated with $K_2SO_4$ (V3) and KCl (V6) had the highest content of tannins (0.34 mg/100 g), followed by the mix (V8) composed by $K_2SO_4$ + $Ca(NO_3)_2$ + KCl (0.33 mg/100 g). In the mix where $KNO_3$ and $KH_2PO_4$ were used (V7), the tannin content was significantly lower as compared with that registered for V3, V6, or V8. The analysis of the electrical signals' values of the treatments for which the highest and the lower content of tannins were registered showed that for all of them, the amplitude of the signals during the fruit development stage was the lowest as compared with the growth stage 2 (BBCH 201–259) and flowering (BBCH 601–605) (Figure 4).

There are many factors that can influence the tannin content; among them, the nutrient availability plays an important role. For instance, it was found that a soil poor in nutrients or with a high C availability determines an increase in the tannin concentrations of plants [78]. The results registered in our experiment showed that the plants either were treated with $KNO_3$, $KH_2PO_4$, or with the mix of all the MaEs (V9), the tannin content was the same. Signs of nutrient stresses were seen for the rest of the treatments where the tannin content was significantly higher.

Oxalate represents another antinutritional factor that when consumed regularly in excessive amounts can lead to nutritional deficiencies or several irritations to the gut's lining. Oxalate forms insoluble salts with $Ca^{2+}$, $Fe^{2+}$, and $Mg^{2+}$, making these minerals unavailable for human and animal nutrition. Moreover, calcium oxalate can precipitate in the kidneys or in the urinary tract, forming crystals that play an important role in the development of kidney stones [76,79]. As in the case of tannins, the concentration of oxalates can vary in plants depending on the specie. Therefore, in legumes (beans, soybean, lentils, etc.), the amount of oxalate can reach 8 mg/kg, in grains (wheat, barley, rye etc.), it can be between 35 and 270 mg/100 g, or in tubers (carrot, sweet potato etc.), it can be between 0.4 and 2.3 mg/100 g [77]. Tomato fruits and especially its derivatives (tomato sauce) are known as a rich source of oxalate [80]. In our experiment, the oxalate content of tomato fruits ranged from 0.14 g/100 g for the plants treated with $KNO_3$ (V2) to 0.20 g/100 g for the control (V11). The amount of oxalate found in the tomato fruits of the plants treated with $KNO_3$ was significantly lower than that found in the plants treated with the mix usually used in agriculture (V9). A high oxalate concentration, namely 0.18 g/100 g, was found in the tomato fruits of the plants treated every day with a different MaEs (V10) or with the mix composed from $K_2SO_4$ + $Ca(NO_3)_2$ + KCl (V8). For these treatments, it was observed that the electrical signals registered for the fruit development stage were different: for V10, the amplitude was higher than that for growth stage 2 (BBCH 201–259) but lower than that of the flowering stage, while for V8, it was the highest as compared with the other two stages considered

(Figure 4). A reason for the high concentration of oxalate in the tomato fruits of the plants treated with V8 and V10 treatments might be because of the interactions between N×K, which enhance the accumulation of this compound [81]. The molar ratio between oxalate and calcium was calculated in order to see to what extent the oxalate affected the calcium bioavailability depending on the treatment applied. The results are shown in Table 4. For a good calcium bioavailability, the ratio should be, according to the WHO, above 2.5. In our experiment, the recorded ratios were bigger than the critical value, ranging between 119.20 mg/100 g (V1 = MgSO4) and 148.80 mg/100 g (V11 = control), suggesting a very low availability of Ca for human nutrition. Anyway, it was observed that in the case of the plants treated with the mix of all the MaEs (V9), the [OX]/[Ca] ratio was increased by 19.35% than that registered for the plants treated with MgSO$_4$ (V1) or by 16.11% for the plants treated with K$_2$SO$_4$, for which we were recorded the highest nutritional values and lycopene content. Therefore, the results suggest that the tomato plants from cv. Brillante F1 do not need all the MaEs at once in order to have high-quality fruits.

Saponins are a group of compounds that can have beneficial effects in humans such as reducing the risk of heart diseases, lowering the cholesterol values, stimulating the immunity, preventing peptic ulcers, or having anticarcinogenic properties, but at the same time, they can reduce the protein digestion or the uptake of vitamins and minerals [76,77]. In tomatoes, the amount of saponins can be between 0.16 and 0.25 mg/100 g [77]. In the tomato fruits of cv. Brillante F1, the saponin content varied with the treatments, the lowest value being registered for V2–KNO$_3$ (0.16 mg/100 g) and the highest for V11–control (0.20 mg/100 g). No significant differences were registered between V11 and V3 (K$_2$SO$_4$), V6 (KCl) and V8 (K$_2$SO$_4$ + Ca(NO$_3$)$_2$ + KCl), for which the content was of 0.19 mg/100 g. Regarding the electrical signals registered, the treatments' signal trend of V11 differs from that of V3 treatment, and all of them were different as compared to that of V3 and V6 treatments (Figure 4).

Alpha-amylase inhibitors are substances that can reduce the carbohydrate digestion and produce various digestive diseases when present in the human diet [82]. In our experiment, the content of $\alpha$-amylase inhibitor varied between 7.51 (V1) and 7.76 mg/mL (V11). The analysis of the electrical signals showed that the amplitude registered for the fruit development stage for V1 treatment was the highest as compared with the growth stage 2 and flowering, while for V11, it was almost the same as the two other phenological stages considered. No significant differences were registered between nine of the treatments (V2–V10).

The last antinutrient factor analyzed was the trypsin's inhibitor, which is known to inhibit the activity of trypsin and chymotrypsin in the gut, impairing the protein digestion. Trypsin inhibitors are widely distributed in *Fabaceae*, *Solanaceae*, and *Gramineae* families [76,77]. The content of trypsin inhibitor in the tomato fruits of cv. Brillante F1 varied depending on the fertilizer applied between 12.35 TUI/mg and 12.77 TUI/mg. The lowest content was registered in the tomato fruits of the plants treated with MgSO$_4$ (12.35 TUI/mg) followed by those treated with KNO$_3$ (12.50 TUI/mg) and K$_2$SO$_4$ (12.63 TUI/mg). For MgSO$_4$ and K$_2$SO$_4$ treatments, it was observed that the trend of the signals registered for the fruit development stage was the same (the lowest as compared to the electrical signals of growth stage 2 and flowering) while for KNO$_3$, it was different (the highest as compared to the electrical signals of the other two phenological stages considered) (Figure 4). The content recorded for these treatments was significantly lower compared with that registered for the tomato fruits of the plants treated with the mix of MaEs (V9).

Considering the results, it can be assumed that high-quality tomato fruits can be obtained by using K$_2$SO$_4$ fertilization instead of the mix commonly used in agriculture (V9).

## 4. Conclusions

The study of the tomato plant cv. Brillante F1 performances under the treatment of different chemical fertilizer monitored by electrical signals showed that depending on the nutrient applied and the plant phenological stages, the shape and the amplitude of the electrical signals were different. The highest signal recorded among the MaEs applied was that generated by Ca(NO$_3$)$_2$. Five different

trends of electric signals were identified during three plant phenological stages (growth stage 2 (BBCH 201–259)—R1, flowering (BBCH 601–605)—R2, and development of fruit (BBCH 701–705)—R3, depending on the MaEs or the mix of MaEs applied. In addition, it was observed that the shape of the signals varied during the day in accordance with the photosynthesis and the amount of $CO_2$ registered. Regarding the nutritional composition and the lycopene content of tomato fruits, it was observed that the highest values were recorded for the plants treated with $K_2SO_4$ (V3), for which the electrical signal amplitude recorded during the development of the fruit stage was the lowest as compared with growth stage 2 and flowering stage. The improvement of fruit quality depended also on the potassium fertilizer forms. Therefore, $K_2SO_4$ fertilization gave a better quality than KCl or $KNO_3$. The electrical signal value for $K_2SO_4$ treatment was the lowest during the fruit development stage compared to the signals registered for growth stage 2 and flowering, while for KCl and $KNO_3$, they were the highest. The analysis on the antinutritional composition of tomato fruits showed that plants treated with $KNO_3$ (V2), for which the electrical signal recorded during the fruit development stage was the highest as compared to growth stage 2 and the flowering stage, had the lowest amount of phytic acid, tannins, oxalate, and saponins, while those treated with $MgSO_4$ (V1), for which the lowest electrical signal was registered during the fruit development stage as compared to growth stage 2 and the flowering stage, had the lowest content of alpha-amylase and trypsin inhibitors. Regarding the bioavailability of zinc, calcium, and iron in the tomato fruits of cv. Brillante F1 for the human diet, regardless of the treatment, phytic acid did not affect the availability of zinc, which was registered as moderate, nor the calcium bioavailability, whose ratio indicated a favorable absorption, but it had a negative impact on iron availability, the ratio recorded being bigger than the critical value. Even though phytic acid did not impair the calcium availability, the molar ratio between oxalate and calcium suggested a very low availability of Ca for human nutrition. However, the values obtained for $K_2SO_4$ fertilization were better than those obtained for the mix commonly used in agriculture. Further studies are needed in order to develop a fertigation scheme based on a smart nutrient use that provides an improved nutritional composition and mineral bioavailability. In addition, it is necessary to evaluate the influence of treatments on yield.

**Author Contributions:** Conceptualization, V.S. and V.M., methodology, V.S. and G.M.; software, M.V.G., G.C.T. and A.C.; validation, V.S., I.B., M.B.; investigation, G.M., C.I.P., I.B., V.A., G.M.; resources, V.S., V.M., data curation, G.M., A.C.; writing—original draft preparation, G.M., M.B.; writing—review and editing, G.M., V.S. and M.B.; supervision, V.S. All authors have read and agreed to the published version of the manuscript.

**Funding:** This research was supported by the project PN-III-P1-1.2-PCCDI-2017-0560 (no. 41/2018) financed by UEFISCDI.

**Conflicts of Interest:** The authors declare no conflict of interest.

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
