# Peer review of "Tomato Crop Performances under Chemical Nutrients Monitored by Electric Signal"

_agronomy, doi:10.3390/agronomy10121915_

Round 1

Reviewer 1 Report

Article
Tomato crop performances under chemical nutrients monitored by electric signal

A brief summary
The article describes reaction on different macronutrients treatment expressed by lycopene, lipid, protein, fibre, dietary fiber, carbotydrates, tannin, oxalate, saponin, a-amylase and trypsin inhibitor content in fruits. The title fits the presented problems. It is pretty well designed, logically constructed and presented paper, appropriate for scientific journal. However, a few corrections and additions can be made.

Specific comments

Line 24: In my opinion keywords should not contain phrases used in the title. Instead of “electrical signal”, “tomato plants”, I propose: “macronutrients”, “lycopene”.

Line 164: Where can ash content be found?

Figure 1: The drawing is a bit messy and contrasts with the nice content of the article. What electrodes were used? Please explain how the electric circuit was grounded. Was the substrate in which the plants grew well insulated from the ground? Was there a situation where water leaking from the pots could close and ground the circuit?

Figure 1 and 2: Figure 2 shows the same signal as in Figure 1. Is it possible to put the descriptions from Figure 2 on Figure 1?

Figure 3. Why the presented data does not cover the whole day (24 hours)?

Figure 1-4 vs. 5: Why does charts on Figure 5 starts at 0:00 and charts on Figures 1-4 at 10:00? Why is the signal for MgSO4 the same in all charts anyway?

Table 2. How was energy calculated (last column)?

Author Response

Answer to Reviewer 1 (red)

Comment 1: Line 24: In my opinion keywords should not contain phrases used in the title. Instead of “electrical signal”, “tomato plants”, I propose: “macronutrients”, “lycopene”.

Answer to Comment 1: “electrical signal”, “tomato plants” were replaced as suggested with “macronutrients”, “lycopene” (Line 48).

Comment 2: Line 164: Where can ash content be found?

Answer to Comment 2:  Ash content was added to Table 2 (Line 385-387).

Also, in the chapter 3.4. Influence of the treatments on the nutritional composition of tomato fruits, additional comments on the ash content were added as follows:

Lines 359 – 361: “Regarding the ash content, it was observed that for K2SO4 (V3) and KCl (V6) treatments the value registered were among the lowest (0.58% – V3; 0.61% – V6), while for V9 treatment it was the highest (0.74%).”

Comment 3: Figure 1: The drawing is a bit messy and contrasts with the nice content of the article. What electrodes were used? Please explain how the electric circuit was grounded. Was the substrate in which the plants grew well insulated from the ground? Was there a situation where water leaking from the pots could close and ground the circuit?

Answer to Comment 3:

Answer to Figure 1: The drawing is a bit messy and contrasts with the nice content of the article. As suggested, to be clearer, the drawing was improved (Line 151-153).

Answer to What electrodes were used? Details about the electrodes were added in the chapter 2.3. Electrical conductivity measurements of the nutrients, Lines 136-138: “The measurements of the electrical signals were performed using a set of stainless-steel electrodes due to their biocompatible proprieties but also to avoid electrochemical reaction between electrodes and ions.”

Answer to Please explain how the electric circuit was grounded. Details about the way the electric circuit was grounded were added in the chapter 2.3. Electrical conductivity measurements of the nutrients, Lines 140– 142: “To eliminate the electrical noises generated by internal and external factors into the electrical circuit used for measurements, the electrical circuit was grounded by connecting to the standard grounding socket of the power supply system.”

Answer to Was the substrate in which the plants grew well insulated from the ground? As mentioned in 2.2. Experimental design, the substrate (Kekkila peat) was in plastic pots with pot plates, therefore it was insulated from the ground.

Answer to Was there a situation where water leaking from the pots could close and ground the circuit? No, we did not meet this situation.

Comment 4: Figure 1 and 2: Figure 2 shows the same signal as in Figure 1. Is it possible to put the descriptions from Figure 2 on Figure 1?

Answer to Comment 4: I supposed your comment is about Figure 2 and 3. To avoid the confusion Figure 3 was deleted, and the details from it was inserted on Figure 2. Also the description of Figure 2 was changed as suggested: “The unique daily imprint of the electrical signal generated by MgSO4 (V1), KNO3 (V2), K2SO4 (V3), Ca(NO3)2 (V4), KH2PO4 (V5), KCl (V6) and H2O (V11 – control) in the tomato plants during 24h.” (Lines 274-277).

As a consequence of deleting Figure 3 the following changes have been made:

Line 274-277: 3 from Figure 3 was replaced with 2 and became Figure 2

Lines 297 – 303: 4 from Figure 4 was replaced with 3 and became Figure 3;

Lines 328 – 349, 343: 5 from Figure 5 was changed to 4 and became Figure 4;

Line 441-444: 6 from Figure 6 was changed to 5 and became Figure 5.

Comment 5: Figure 3. Why the presented data does not cover the whole day (24 hours)?

Answer to Comment 5: Taking into account the previous comment, Figure 3 was deleted in order to avoid the confunsion and redudancy. Information presented in Figure 3 were inserted in Figure 2.

Comment 6: Figure 1-4 vs. 5: Why does charts on Figure 5 starts at 0:00 and charts on Figures 1-4 at 10:00? Why is the signal for MgSO4 the same in all charts anyway?

Answer to Comment 6: To be clearer charts on Figure 5 (now Figure 4) were enhanced and replaced, also Figure 4 (now Figure 3).

The charts on Figure 4 (now Figure 3) starts at 10 and end at 20:30 because the photosynthetic rate and sub-stomatal CO2 measurements were done during that interval of time.

Comment 7: Table 2. How was energy calculated (last column)?

Answer to Comment 7: The energy was calculated according to the REGULATION (EU) No 1169/2011 OF THE EUROPEAN PARLIAMENT AND OF THE COUNCIL of 25 October 2011 on the provision of food information to consumers, amending Regulations (EC) No 1924/2006 and (EC) No 1925/2006 of the European Parliament and of the Council, and repealing Commission Directive 87/250/EEC, Council Directive 90/496/EEC, Commission Directive 1999/10/EC, Directive 2000/13/EC of the European Parliament and of the Council, Commission Directives 2002/67/EC and 2008/5/EC and Commission Regulation (EC) No 608/2004,

using the suggested conversion factors as follows: Energy value (kcal) = Proteins (g) x 4kcal / g + Carbohydrates (g) x 4kcal / g + Lipids (g) x 9kcal / g + Fibers (g) x 2kcal / g. (Lines 170-172)

Reviewer 2 Report

This work is analyzing the electrical signals of the nutrients applied individually or in different mixes, correlating the electrical signals with the leaf gas exchange processes, studying the relation between the electrical signals and different plant phenological stages. This part is innovative and very interesting. However, then the authors provide results of the influence of the different nutrient treatments on nutritional composition of tomato fruits and antinutritional factors in fruits which are not related with the electrical signals. It seems rather like the authors connected two different papers. Besides, the second part is already studied and it is well known long time ago that plants need mainly fertilizers based on potassium for a high quality of the fruits and that the improvement of fruit quality depends on the potassium fertilizer forms.

The results would be sufficient only with the first part if the authors had included the influence of the different nutrient treatments on tomato yield. However, they state that it was not evaluated in this study. Furthermore, it would be interesting to provide an economic assessment of using different nutrient treatments related to the production cost.

Author Response

Answer to Reviewer 2 (orange)

Comment 1: However, then the authors provide results of the influence of the different nutrient treatments on nutritional composition of tomato fruits and antinutritional factors in fruits which are not related with the electrical signals. 

Answer to Comment 1: As suggested the influence of the different nutrient treatments on nutritional composition of tomato fruits and antinutritional factors in fruits were related with the electrical signals as follows:

Lines 362 – 369: “The analysis of the electrical signals trend recorded during the development of fruit stage (BBCH 701–705) showed that for V3 treatment it was registered the lowest electrical signal as compared with growth stage 2 (BBCH 201–259) and flowering stage (BBCH 601–605), for V6 it was recorded the highest electrical signal as compared with the other two stages while for the mix usually used in agriculture (V9), the electrical signal was higher than that for growth stage 2 (BBCH 201–259), but lower than that of the flowering stage. The values of the electrical signals registered for V3 varied between 3.91x10-5 A and 4.96x10-5 A, for V6 between 1.09x10-4 A and 1.23x10-4 A and for V9 between 6.00x10-5 A and 7.04x10-5 A (Figure 4c).”

Lines 414 – 417: “As mentioned before, the electrical signals’ trends recorded for this treatment during the development of fruit stage (BBCH 701–705) was the lowest as compared with growth stage 2 (BBCH 201–259) and flowering stage (BBCH 601–605). The value of the electrical signal varied between 3.91x10-5 A and 4.96x10-5 A (Figure 4c).”

Lines 419 – 421: “For both of these treatments the electrical signals’ trend recorded were the highest during the fruit development stage (BBCH 701–705), as compared with growth stage 2 (BBCH 201–259) and flowering stage (BBCH 601–605) (Figures 4b and 4h).”

Lines 425 – 427: “The electrical signals recorded for both V9 and V10 treatments showed that during the fruit development stage (BBCH 701–705) the amplitude of the signals was higher than that for growth stage 2 (BBCH 201–259), but lower than that of the flowering stage (Figures 4i and 4j).”

Lines 432 – 440: “The analysis of the electrical signals’ amplitude showed that for K2SO4 fertilizer the registered value varied between 3.91x10-5 A and 4.96x10-5 A, for KNO3 between 4.43x10-5 A and 6.53x10-5 A and for KCl between 1.09x10-4 A and 1.23x10-4 A (Figure 4c). Also, it was interesting to observe that for lycopene, as for the nutritional composition, the highest values were registered for treatments whose electrical signals during the fruit development stage were the lowest or the highest, while the lowest values were for those which amplitude of the signals was higher than that for growth stage 2 (BBCH 201–259), but lower than that of the flowering stage.”

Lines 482 – 484: “The electrical signals recorded for all these treatments during the development of fruit stage (BBCH 701 – 705) were the highest as compared to growth stage 2 (BBCH 201–259) and flowering stage (BBCH 601–605) (Figure 4).”

Lines 496 – 498: “…whose electrical signal during the fruit development stage was higher than that for growth stage 2 (BBCH 201–259), but lower than that of the flowering stage and value varied between 1.03x10-4 A and 7.26x10-5A (Figure 4).”

Lines 499 – 501: “For this, the electrical signal value during the fruit development stage was the lowest as compared with the other two stages taken into consideration and varied between 3.43x10-5 A and 4.97x10-5 A.”

Lines 550 – 553: “The analysis of the electrical signals’ values of the treatments for which the highest and the lower content of tannins were registered, showed that for all of them the amplitude of the signals during the fruit development stage was the lowest as compared with the growth stage 2 (BBCH 201–259) and flowering (BBCH 601–605). The differences were done by the values registered (Figure 4).”

Lines 574 -577: “For these treatments it was observed that the electrical signals registered for the fruit development stage were different: for V10 the amplitude was higher than that for growth stage 2 (BBCH 201–259), but lower than that of the flowering stage, while for V8, was the highest as compared with the other two stages considered (Figure 4).”

Lines 599 – 600: “Regarding the electrical signals registered, the treatments’ signal trend of V11 differ on that of V3 treatment and all of them were different as compared to that of V3 and V6 treatments (Figure 4).”

Lines 603 – 606: “The analysis of the electrical signals showed that the amplitude registered for the fruit development stage for V1 treatment was the highest as compared with the growth stage 2 and flowering, while for V11 was almost the same as the two other phenological stages considered.”

Lines 614 – 617: “For MgSO4 and K2SO4 treatments, it was observed that the trend of the signals registered for the fruit development stage was the same (the lowest as compared to the electrical signals of growth stage 2 and flowering) while for KNO3 was different (the highest as compared to the electrical signals of the other two phenological stages considered) (Figure 4).”

Comment 2:  It would be interesting to provide an economic assessment of using different nutrient treatments related to the production cost. 

Answer to Comment 2: The results presented in this study represent the first step in a series of experiments that will follow, which will aim to obtain a fertigation scheme based on a smart nutrient use. Therefore, since this is not the final scheme, we did not manage to provide an economic assessment of using different nutrient treatments related to the cost production to not create confusion. Also, we would like to improve the scheme taking into account other parameters, like production, to have a clearer image. At that point we will make an economic assessment which in our opinion will be more genuine.

Reviewer 3 Report

The experiment is well contacted. The methodology is sound and the presentation is clear. 

The results are very interesting, but not connected between them. The conclusions should be rewritten pointing out more clearly the connection between the results. 

Author Response

Answer to Reviewer 3 (underline blue)

Comment 1: The results are very interesting, but not connected between them.

Answer to Comment 1: Connections between the influence of the different nutrient treatments on nutritional composition of tomato fruits, antinutritional factors in fruits and electrical signals were done as follows:

Lines 362 – 369: “The analysis of the electrical signals trend recorded during the development of fruit stage (BBCH 701–705) showed that for V3 treatment it was registered the lowest electrical signal as compared with growth stage 2 (BBCH 201–259) and flowering stage (BBCH 601–605), for V6 it was recorded the highest electrical signal as compared with the other two stages while for the mix usually used in agriculture (V9), the electrical signal was higher than that for growth stage 2 (BBCH 201–259), but lower than that of the flowering stage. The values of the electrical signals registered for V3 varied between 3.91x10-5 A and 4.96x10-5 A, for V6 between 1.09x10-4 A and 1.23x10-4 A and for V9 between 6.00x10-5 A and 7.04x10-5 A (Figure 4c).”

Lines 414 – 417: “As mentioned before, the electrical signals’ trends recorded for this treatment during the development of fruit stage (BBCH 701–705) was the lowest as compared with growth stage 2 (BBCH 201–259) and flowering stage (BBCH 601–605). The value of the electrical signal varied between 3.91x10-5 A and 4.96x10-5 A (Figure 4c).”

Lines 419 – 421: “For both of these treatments the electrical signals’ trend recorded were the highest during the fruit development stage (BBCH 701–705), as compared with growth stage 2 (BBCH 201–259) and flowering stage (BBCH 601–605) (Figures 4b and 4h).”

Lines 425 – 427: “The electrical signals recorded for both V9 and V10 treatments showed that during the fruit development stage (BBCH 701–705) the amplitude of the signals was higher than that for growth stage 2 (BBCH 201–259), but lower than that of the flowering stage (Figures 4i and 4j).”

Lines 432 – 440: “The analysis of the electrical signals’ amplitude showed that for K2SO4 fertilizer the registered value varied between 3.91x10-5 A and 4.96x10-5 A, for KNO3 between 4.43x10-5 A and 6.53x10-5 A and for KCl between 1.09x10-4 A and 1.23x10-4 A (Figure 4c). Also, it was interesting to observe that for lycopene, as for the nutritional composition, the highest values were registered for treatments whose electrical signals during the fruit development stage were the lowest or the highest, while the lowest values were for those which amplitude of the signals was higher than that for growth stage 2 (BBCH 201–259), but lower than that of the flowering stage.”

Lines 482 – 484: “The electrical signals recorded for all these treatments during the development of fruit stage (BBCH 701 – 705) were the highest as compared to growth stage 2 (BBCH 201–259) and flowering stage (BBCH 601–605) (Figure 4).”

Lines 496 – 498: “…whose electrical signal during the fruit development stage was higher than that for growth stage 2 (BBCH 201–259), but lower than that of the flowering stage and value varied between 1.03x10-4 A and 7.26x10-5A (Figure 4).”

Lines 499 – 501: “For this, the electrical signal value during the fruit development stage was the lowest as compared with the other two stages taken into consideration and varied between 3.43x10-5 A and 4.97x10-5 A.”

Lines 550 – 553: “The analysis of the electrical signals’ values of the treatments for which the highest and the lower content of tannins were registered, showed that for all of them the amplitude of the signals during the fruit development stage was the lowest as compared with the growth stage 2 (BBCH 201–259) and flowering (BBCH 601–605). The differences were done by the values registered (Figure 4).”

Lines 574 -577: “For these treatments it was observed that the electrical signals registered for the fruit development stage were different: for V10 the amplitude was higher than that for growth stage 2 (BBCH 201–259), but lower than that of the flowering stage, while for V8, was the highest as compared with the other two stages considered (Figure 4).”

Lines 599 – 600: “Regarding the electrical signals registered, the treatments’ signal trend of V11 differ on that of V3 treatment and all of them were different as compared to that of V3 and V6 treatments (Figure 4).”

Lines 603 – 606: “The analysis of the electrical signals showed that the amplitude registered for the fruit development stage for V1 treatment was the highest as compared with the growth stage 2 and flowering, while for V11 was almost the same as the two other phenological stages considered.”

Lines 614 – 617: “For MgSO4 and K2SO4 treatments, it was observed that the trend of the signals registered for the fruit development stage was the same (the lowest as compared to the electrical signals of growth stage 2 and flowering) while for KNO3 was different (the highest as compared to the electrical signals of the other two phenological stages considered) (Figure 4).”

Comment 2: The conclusions should be rewritten pointing out more clearly the connection between the results. 

Answer to Comment 2: The conclusions were supplemented with information as suggested by the reviewer as follows:

Lines 634 – 638: “….for which the electrical signal amplitude recorded during the development of fruit stage (BBCH 701–705) was the lowest as compared with growth stage 2 (BBCH 201–259) and flowering stage (BBCH 601–605). The values recorded for the mix usually used in agriculture (V9) were significantly lower than that of K2SO4 (V3), the amplitude of the signals during the fruit development stage being higher than that of growth stage 2, but lower than that of the flowering stage.”

Lines 640 – 642: “The electrical signal value for K2SO4 treatment was the lowest during the fruit development stage compared to the signals registered for growth stage 2 and flowering, while for KCl and KNO3 were the highest”

Round 2

Reviewer 2 Report

The paper is improved as the authors added discussion relating the different nutrient treatments on nutritional composition of tomato fruits and antinutritional factors in fruits with the electrical signals.

I could not find though where this relation is concluding in the Conclusions section. E.g. in lines 634-638 and 640-642 the authors provide again results which should move to the section Results and Discussion. They should instead give clear conclusions at the end.

Also, in lines 105-106 the authors write that they are correlating the electrical signals only with the photosynthetic processes but not with different plant phenological stages or nutritional composition of tomato fruits and antinutritional factors in fruits.

I believe the paper should be improved by revising the Conclusions section and adding the correlation between electrical signals and different plant phenological stages or nutritional composition of tomato fruits and antinutritional factors in fruits signals.

Author Response

Dear Editor,

We thank you again for your interest in our paper. We also thank the reviewer for the patient and careful examination of our manuscript and for providing ideas and corrections that will improve this manuscript. Our point-by-point responses regarding comments are detailed on the following pages. All the changes suggested by Reviewer 2 were highlighted with orange. Other changes were done with green.

Answer to Reviewer 2 (orange)

Comment 1: The paper is improved as the authors added discussion relating the different nutrient treatments on nutritional composition of tomato fruits and antinutritional factors in fruits with the electrical signals. I could not find though where this relation is concluding in the Conclusions section. E.g. in lines 634-638 and 640-642 the authors provide again results which should move to the section Results and Discussion. They should instead give clear conclusions at the end.

Answer to Comment 1: Lines 634-638 and lines 640-642 were deleted. As suggested by the reviewer, concluding remarks regarding the relation between parameters analysed were added as follows:

Lines 627 – 630: “The study of the tomato plants cv. Brillante F1 performances under the treatment of different chemical fertilizer monitored by electrical signals showed that depending on the nutrient applied and the plant phenological stages, the shape and the amplitude of the electrical signals were different.”

Lines 635 – 638:  “Regarding the nutritional composition and the lycopene content of tomato fruits it was observed that the highest values were recorded for the plants treated with K2SO4 (V3) for which the electrical signal amplitude recorded during the development of fruit stage was the lowest as compared with growth stage 2 and flowering stage.”

Lines 642 – 648: “The analysis on the antinutritional composition of tomato fruits showed that plants treated with KNO3 (V2), for which the electrical signal recorded during the fruit development stage was the highest as compared to growth stage 2 and flowering stage, had the lowest amount of phytic acid, tannins, oxalate and saponins, while those treated with MgSO4 (V1), for which was registered the lowest electrical signal during the fruit development stage as compared to growth stage 2 and flowering stage, had the lowest content of alpha–amylase and trypsin inhibitors.”

Comment 2:  Also, in lines 105-106 the authors write that they are correlating the electrical signals only with the photosynthetic processes but not with different plant phenological stages or nutritional composition of tomato fruits and antinutritional factors in fruits.

Answer to Comment 2: As suggested by the reviewer the following changes were done:

Line 104 – 106: “This study is trying to answer these questions, by analyzing the electrical signals of the nutrients applied individually or in different mixes, correlating the electrical signals with the leaf gas exchange processes, studying the relation between the electrical signals and different plant phenological stages and the influence of the treatments on the lycopene content, nutritional and antinutrient composition, also the mineral bioavailability of tomato fruits cv. Brillante F1.”

was changed to (Lines 104 – 109):

“This study is trying to answer these questions, by analyzing the electrical signals of the nutrients applied individually or in different mixes, by studying the influence of the treatments on the lycopene content, nutritional and antinutrient composition, also the mineral bioavailability of tomato fruits cv. Brillante F1 and by correlating the electrical signals with the leaf gas exchange processes, different plant phenological stages, lycopene content, nutritional and antinutrient composition.”

Comment 3:  I believe the paper should be improved by revising the Conclusions section and adding the correlation between electrical signals and different plant phenological stages or nutritional composition of tomato fruits and antinutritional factors in fruits signals.

Answer to Comment 3: As suggested by the reviewer the Conclusions section was improved by adding the correlation between electrical signals, nutrient applied, different plant phenological stages and leaf gas exchange processes (Lines 627 – 635): “The study of the tomato plants cv. Brillante F1 performances under the treatment of different chemical fertilizer monitored by electrical signals showed that depending on the nutrient applied and the plant phenological stages, the shape and the amplitude of the electrical signals were different. The highest signal recorded among the MaEs applied was that generated by Ca(NO3)2. Five different trends of electric signals were identified during three plant phenological stages (growth stage 2 (BBCH 201–259) – R1, flowering (BBCH 601–605) – R2 and development of fruit (BBCH 701–705) – R3) depending on the MaEs or the mix of MaEs applied. Also, it was observed that the shape of the signals varied during the day in accordance with the photosynthesis and the amount of CO2 registered.”

Also, concluding remarks regarding the correlations between nutritional composition, lycopene content and electrical signals were added (Lines 635 – 638): “Regarding the nutritional composition and the lycopene content of tomato fruits it was observed that the highest values were recorded for the plants treated with K2SO4 (V3) for which the electrical signal amplitude recorded during the development of fruit stage was the lowest as compared with growth stage 2 and flowering stage.”

Nevertheless, concluding remarks regarding the correlation between electrical signals and antinutritional factors were added (Lines 642 – 648): “The analysis on the antinutritional composition of tomato fruits showed that plants treated with KNO3 (V2), for which the electrical signal recorded during the fruit development stage was the highest as compared to growth stage 2 and flowering stage, had the lowest amount of phytic acid, tannins, oxalate and saponins, while those treated with MgSO4 (V1), for which was registered the lowest electrical signal during the fruit development stage as compared to growth stage 2 and flowering stage, had the lowest content of alpha–amylase and trypsin inhibitors.”

Other changes (green)

Line 155: “the stomatal conductance (gs, μmol H2O m–2 s–1” from ‚’’The photosynthetic rate (µmol CO2 m–2 s–1), the stomatal conductance (gs, μmol H2O m–2 s–1)….” was deleted.

Line 288: “photosynthetic process” was replaced with “leaf gas exchange processes”

Line 291: stomatal conductance from “(photosynthesis rate, stomatal conductance...” was deleted

This manuscript is a resubmission of an earlier submission. The following is a list of the peer review reports and author responses from that submission.